# Incidence of Upper Body Injuries in Amateur Padel Players

**DOI:** 10.3390/ijerph192416858

**Published:** 2022-12-15

**Authors:** Diego Muñoz, Manuel Coronado, María C. Robles-Gil, Manuel Martín, Adrián Escudero-Tena

**Affiliations:** 1Department of Musical, Plastic and Corporal Expression, Faculty of Sport Sciences, University of Extremadura, 10003 Caceres, Spain; 2Padel MBA, 28729 Madrid, Spain; 3Training Optimization and Sport Performance Research Group (GOERD), Sport Science Faculty, University of Extremadura, 10005 Caceres, Spain

**Keywords:** racket sports, health, prevention, sex, rackets

## Abstract

The objectives of this study were to analyze the injuries suffered during the previous year by amateur padel players according to the characteristics of the racket, their usual volume of practice and their experience in padel. A total of 950 amateur players (*X* age: 31.68 years; *X* weight: 70.84 kg; *X* height: 170.9 cm) participated voluntarily, completing an ad-hoc questionnaire. The results indicated that the appearance of the injuries and their location was different according to the sex of the amateur padel players. Men had a higher incidence of muscle and ligament injuries in the shoulder, and tendon injuries in the elbow. On the other hand, women had a greater probability of having muscle injuries in the shoulder and arm, ligament injuries in the elbow and bone injuries in the wrist and elbow. In general, tendon injuries were the most common injury in padel and the shoulder and elbow were the most affected areas. Moreover, men tend to use heavy (CSR = 6.0), fiberglass or carbon (CSR = 2.1), diamond-shaped rackets (CSR = 3.2), with a hard core (CSR = 4.4) and with two or more over grips (CSR = 2.7). Women usually use less heavy (CSR = 6.0), round-shaped rackets (CSR = 4.9), with a soft core (CSR = 4.4) and with one or no over grips (CSR = 2.7). In addition, men tend to play padel more often and have been practicing for longer. In conclusion, although the risk of injury depends on many factors, we identified that the characteristics of the racket, the volume of weekly practice, the experience of the player and the gender of the player are fundamental aspects to take into account for the prevention of injuries in amateur padel players.

## 1. Introduction

The number of scientific papers that focus on padel as an object of study has increased in recent years [1] due to the great importance that this sport has acquired, as it is practiced in more than 50 countries. A parallel increase has been seen in the number of facilities, professional circuits (World Padel Tour, APT padel tour or Premier Padel), commercial agreements (sponsorships, employment contracts, etc.) and sport licenses. [2]. Research topics in padel address a large number of areas, including game analysis [3,4,5], physiology [6], anthropometric profiles [7], psychology [8], education [9] and biomechanics [10].

Likewise, scientific investigations exploring sport injuries in padel have been carried out [11,12,13,14]. The risk of injury is inherent in any sporting activity due to its nature, but it is possible to reduce this risk if greater knowledge about it and how to carry out adequate preventive work is achieved. Although injuries have a multifactorial origin, there are some known risk factors, which may be specific to the sporting activity.

In padel, the lower extremities are the areas that are most affected by injury, followed by the upper extremities, the trunk and, finally, the head and neck [13]. Regarding the exact anatomical location, the elbow is the most affected, and tendon-type injuries stand out (followed by muscular ones), epicondylitis being the most frequent injury among padel players [13,15]. In addition, other injuries are frequent, such as ankle sprains due to lateral movements [16], internal calf muscle injuries, knee ligament injuries, acute lumbosciatica, rotator cuff tendinitis, scaphoid fracture and ocular injuries [15]. The relationship between injuries and variables, such as gender, age and the level of the players, has also been analyzed, with the latter factor standing out because it seems that a greater number of injuries occur in lower-level players [12]. It is possible that the lesser physical condition and the worse technical execution [17] of lower-level players are responsible for this type of injury, which indicates the importance of physical preparation and correct technical execution in the prevention of injuries.

To date, no studies have attempted to analyze the relationship between certain elements of padel (type of surface, facility materials, type of racket, type of ball, etc.) and the incidence of injuries in padel players, which could be a great step forward for the prevention of injuries in padel players. In addition, there is an increasingly greater variety of the elements mentioned above on offer in the market. Specifically, the padel racket is a fundamental element in the game, and it can directly influence the player’s upper body. There is a wide variety of brands and models on the market, with many different variables (shape, weight, racket thickness, face and core composition materials, etc.) that give each of them a series of features. Therefore, the objectives of this study were: I. to identify the incidence of upper body injuries in amateur padel players, II. to analyze the differences between men’s and women’s amateur padel players according to the most common characteristics of the racket, the experience and volume of habitual practice and the characteristics of the injuries in their upper limbs, III. to determine the relationships between the most frequent injuries among amateur padel players and the characteristics of the racket, the years of experience of the players and their amount of weekly practice and IV. to determine the relationship between the most frequent type of injury and its location in amateur padel players.

## 2. Materials and Methods

### 2.1. Research Design

The design of this research falls under the empirical methodology and, more specifically, it is a study with a descriptive or associative strategy, according to statistical analysis [18].

### 2.2. Sample

This study was approved by the Bioethics Commission of the University of Extremadura (reference 154/2000). A total of 950 amateur padel players (803 men and 147 women) from different locations participated, recording responses from 12 different countries. Spain was the country with the highest response rate (49 different provinces). Table 1 shows the general characteristics of the participants (age, weight and height), segmented. Age is similar, but there are significant differences (*p* < 0.001) in terms of weight and height between the sexes, both being greater in men compared to women.

### 2.3. Study Variables

Taking, as a reference, the questionnaires used in previous investigations [13], the following variables were defined and analyzed based on their categorical core and their degree of openness [19]:-Category: the men’s and women’s categories were established in order to analyze the possible differences between them.-Injury: differentiating between the players who suffered an injury from those who were not injured.-Type of injury: establishing an adaptation of the classification made by Sánchez-Alcaraz and collaborators [12], and differentiating between ligamentous, muscular, tendinous and bone injuries.-Anatomical location of the injury: establishing the locations of the injury, including the shoulder, arm, elbow, forearm, wrist and hand.-Years of experience/padel: differentiating between those who had been practicing padel for less than five years and those who had equaled or exceeded five years of padel practice.-Hours per week/padel: refers to the number of hours that the players practiced padel. Two groups were established (less than three hours/week, and three or more than three hours/week).-Practice of another sport: differentiating between those players who practiced another sport and those who did not.-Racket shape: differentiating between a round racket shape, teardrop shape and diamond shape.-Type of core: a distinction was made between padel rackets with a soft core and a hard core.-Weight of the padel racket (g): refers to the weight (in grams) of the padel racket. Two categories were established (less than 350 g and equal to or more than 350 g).-Number of over grips: deals with the number of over grips that the players used in the first of their rackets. Two categories were established (none and one on one side and two or more on the other).-Racket composition: the fiberglass and carbon fiber categories were established. In addition, the “unknown” category was included in case the players did not know the composition of their padel racket.

### 2.4. Process

Once the questionnaire, specific for this manuscript, had been prepared, the data were collected through Google (Google Forms). The participants filled out the questionnaire voluntarily, after informed consent was obtained in which the confidentiality and anonymity of their responses were ensured, following approval from the Bioethics Committee of the University of Extremadura (reference 154/2020). No researcher was present with the participants at the time they answered the questionnaire.

### 2.5. Statistical Analysis

A descriptive analysis was performed in order to obtain information regarding the number of times the categories of each study variable occurred (frequency and percentage). An inferential analysis was conducted in order to develop contingency tables, including the Chi-square (χ^2^) statistical test in order to obtain the association between variables. The strength of association between the variables was also calculated, for which Cramer’s V coefficient (Vc) was used [20]. Cramer’s V coefficient is widely used in sport science studies. Crewson differentiates the strength of the association based on the value, considering a small (<0.100), low (0.100–0.299), moderate (0.300–0.499) or high (>0.500) association [21]. In addition, subsequent Z-tests were performed in order to compare the column proportions, adjusting *p*-values <0.05 according to Bonferroni. The contingency tables made it possible to identify the associations between the categories of the variables through the corrected standardized residuals (CSR). Residuals > |1.96| reported more or fewer cases than there should be [20]. The statistical analysis was performed using the statistical package SPSS 27.0 for Windows.

## 3. Results

The results indicate that the weight (χ^2^ (1) = 35.984; *p* < 0.001; Vc = 0.195), the type of core (χ^2^ (1) = 19.632; *p* < 0.001; Vc = 0.144), the shape (χ^2^ (2) = 25.677; *p* < 0.001; Vc = 0.164), the number of over grips (χ^2^ (1) = 7.522; *p* = 0.006; Vc = 0.089) and the composition (χ^2^ (2) = 12.750; *p* = 0.002; Vc = 0.116) of the rackets used by amateur padel players is associated with the sex of the players. Thus, Table 2 shows the differences between men’s and women’s amateur players in the characteristics of their padel rackets.

Men tend to use more diamond-shaped rackets, whereas women use more round-shaped rackets. Moreover, the type of core that male amateur padel players usually use is hard, whereas female amateur padel players tend to use soft core rackets more. Men use heavier padel rackets compared to women and tend to put more over grips on the padel grips. Finally, men use carbon fiber rackets more than women, and women are more likely to be unaware of the composition of their rackets.

Gender is associated with the volume of practice “hours/weeks” (χ^2^ (1) = 8.299; *p* = 0.004; Vc = 0.093) and with experience “time practicing (years)” (χ^2^ (1) = 7.135; *p* = 0.008; Vc = 0.087) playing amateur padel. Table 3 shows the game differences between men and women in amateur padel, according to their practice in hours per week and according to their experience in years. Men play more hours of padel per week than women (CSR = 2.9). In addition, men tend to spend more time practicing amateur padel than women (CSR = 2.7).

The results show that gender is not associated with the type of injury (χ^2^ (3) = 5.358; *p* = 0.147; Vc = 0.105) and its location (χ^2^ (5) = 1.838; *p* = 0.871; Vc = 0.062) in amateur padel. Table 4 shows the characteristics of injuries to the upper body of amateur players. It can be seen that tendinous injuries are the most common in padel, followed by muscular injuries. The shoulder and elbow are the specific locations most affected, and there are more ligamentous injuries in women than in men.

The weight of the racket used by amateur players is associated with the appearance of injuries in the upper body (χ^2^ (2) = 5.100; *p* = 0.024; Vc = 0.073). However, other characteristics, such as the type of core (χ^2^ (1) = 2.714; *p* = 0.099; Vc = 0.053), the shape (χ^2^ (2) = 2.773; *p* = 0.250; Vc = 0.054), the number of over grips (χ^2^ (1) = 2.472; *p* = 0.116; Vc = 0.051) and the composition (χ^2^ (2) = 4.517; *p* = 0.105; Vc = 0.105), are not associated with the appearance of injuries in amateur padel players. Table 5 shows the relationships between the appearance of injury according to various characteristics of the padel racket used by amateur players. A weight equal to or greater than 350 g seems to be a parameter that is related to the appearance of injuries in amateur padel players.

The results show that the incidence of injuries is associated with the volume of practice “hours/weeks” (χ^2^ (2) = 10.831; *p* = 0.004; Vc = 0.107) and experience “practicing time (years)” (χ^2^ (1) = 6.895; *p* = 0.009; Vc = 0.085) playing amateur padel. However, practicing another sport does not influence the appearance of padel injuries in the upper body (χ^2^ (1) = 0.250; *p* = 0.617; Vc = 0.016). Table 6 shows the incidence of injuries according to padel practice (in hours per week) and experience (years), in addition to the practice of another sport. Playing padel for more than six hours a week usually leads to the appearance of injuries in amateur padel players, as does having little experience (<5 years).

The type of injury in amateur padel players is associated with its location in men (χ^2^ (15) = 94.423; *p* < 0.001; Vc = 0.273) and in women (χ^2^ (15) = 38.788; *p* = 0.001; Vc = 0.468). Table 7 shows the relationship between the type of injury and its location in amateur padel players.

In men, shoulder injuries are usually muscular and ligamentous. In addition, muscle injuries also usually appear in the forearm, whereas tendon injuries usually appear in the elbow. Finally, bone lesions are characteristic of the wrist and hand. In women, muscle injuries usually appear in the shoulder and arm, whereas ligament injuries usually appear in the elbow and bone injuries usually appear in the wrist and hand.

## 4. Discussion

The objectives were to analyze the characteristics of the rackets of male and female amateur padel players, their volume of habitual practice and their experience playing padel, and how these factors affected the appearance of upper body injuries according to their type and location. An injury is a negative event that can be caused by many factors, whether sport-related, psychological or psychosocial. However, athletes give greater importance to sport-related factors [22]. The results obtained in this study indicate that the characteristics of men’s rackets are different from those used by women. Therefore, it is essential that companies dedicated to the design and manufacturing of rackets take into account the gender of the player as a factor. Although previous research related to the characteristics of padel rackets was not found, several studies indicate that anthropometric and strength characteristics, which are different between men and women (taller men, with a higher percentage of muscle and higher levels of vertical jump and grip strength [23,24]), influence the style of play (men make more shots closer to the net, developing a more aggressive game, whereas women make more shots from the middle and back of the court, developing a more conservative game [25,26,27,28,29,30]) and this, in turn, is directly influenced by the characteristics of the racket. Men use more diamond-shaped carbon fiber rackets, which are heavier, with a hard core and with two or more over grips, whereas women use lighter round-shaped rackets, with a soft core and with one or no over grips. Thus, the importance of choosing a suitable racket according to the athlete’s anthropometric characteristics and style of play, and not to fulfill aesthetic or sponsorship criteria, is fundamental.

Furthermore, in terms of the differences between the sexes of amateur players, men tend to play padel more often (three or more hours of practice per week) and have been practicing it for longer (five or more years of practice experience). Numerous studies [31,32,33] have shown that male subjects have a more positive general attitude compared to female subjects when practicing a physical activity or are more likely to practice physical and sporting activities than the opposite sex. Therefore, the results obtained regarding the usual practice of padel and the amount of time spent practicing seem logical.

Similarly, the appearance of injuries and their location differs according to the sex of amateur padel players. Men who practice padel at the amateur level suffer more injuries of the muscular and ligamentous type in the shoulder, in the forearm muscles, in the elbow tendons and in the wrist and hand bones. However, women usually suffer muscular injuries in the shoulder and arm, ligamentous injuries in the elbow and in the wrist and bone injuries in the hand. Therefore, although there are many intrinsic factors (previous injuries, age, body composition, health status, physical condition, etc.) and extrinsic factors (sport-specific motor skills, training, the quality of technical gestures, materials, the environment, etc.) that determine the appearance of lesions [34], gender is a fundamental intrinsic factor in the appearance of lesions and their characteristics.

Tendinous injuries are the most common in padel, followed by muscular injuries in amateur padel players, and the shoulder and elbow are the specific locations that are most affected. In addition, women tend to suffer more ligament injuries than men, so they should perform more preventive exercises for this type of injury. The results obtained in this study coincide with other similar studies [13,14,15], where tendon injuries were found to be the most numerous and the elbow joint was the most affected in padel. These same studies show that epicondylitis (commonly known as “tennis elbow”) is the most frequent injury in padel. We should also highlight the shoulder joint, which was revealed to be the second most affected, and muscular injuries, which also have a high incidence.

The appearance of injuries is greater in amateur padel players when the weight of their racket is greater than or equal to 350 g and when they play more than six hours a week. However, the appearance of injuries is reduced when players have been practicing padel for more than five years. García-Fernández and collaborators [13] have shown that an increase in the hours spent practicing padel causes sport-related injuries among players. In addition, previous studies among players have determined that their technical level could be related to the appearance of injuries since players at the highest level have fewer injuries [12]. Thus, players must devote time to technical work in the execution of gestures, especially in the initial stages.

Despite the interesting findings found in this research, the limitations of this study should be noted, which must be taken into account when interpreting the results obtained. Firstly, the number of questionnaires completed by women was considerably lower than that of men. Secondly, in addition to the characteristics of the racket, the volume of practice and experience, other factors, such as the level of physical condition, level of technique and fatigue, could be the cause of the injuries and requires further investigation in the future. 

## 5. Conclusions

Tendinous injuries are the most common injuries in padel, followed by muscular injuries. In addition, the shoulder and elbow are the specific locations most affected. Women tend to suffer more ligamentous injuries than men, so they should perform more preventive exercises for this type of injury. In addition, amateur padel players suffer more muscular and ligamentous injuries to the shoulder, forearm muscles, elbow tendons and wrist and hand bones. Amateur players tend to suffer more from muscular injuries in the shoulder and arm, ligamentous injuries in the elbow and bone injuries in the wrist and hand.

Men use heavier, carbon fiber diamond-shaped rackets, with a hard core and two or more over grips, whereas women use lighter, more round-shaped rackets, with a soft core and with one or no over grips. This information is essential for companies that are dedicated to the design and manufacturing of rackets because their characteristics should differ according to gender. Moreover, in terms of the differences between mael and female amateur players, men tend to play padel more often and have been practicing for longer.

Finally, amateur padel players should carry out preventive training because the appearance of injuries in this sport is common. The appearance of injuries increases in padel players when the weight of their racket is greater than or equal to 350 g and when they play more than six hours a week, although it is reduced when players have been practicing padel for more than five years. Therefore, adjusting to the ideal weight of the racket, not abusing practice and having experience is beneficial and prevents the appearance of injuries in padel.

## Figures and Tables

**Table 1 ijerph-19-16858-t001:** General characteristics of the participants.

Variables	Men		Women		*p*
	*X*	*±SD*	*X*	*±SD*	
Age (years)	29.99	±10.80	33.37	±13.03	
Weight (kg)	78.71	±14.53	62.97	±11.33	*
Height (cm)	177.51	±7.25	164.29	±6.47	*

*X* average; SD: standard deviation; * *p* < 0.001 comparing men and women.

**Table 2 ijerph-19-16858-t002:** Differences in the characteristics of the rackets used by men’s and women’s amateur padel players.

Variables		Men			Women		
		N	%	CSR	N	%	CSR
Padel racket shape	Round	196	24.4 a	−4.9 *	52	44.2 b	4.9 *
	Tear	319	39.7 a	1.5	47	33.3 a	−1.5
	Diamond	288	35.9 a	3.2 *	30	22.4 b	−3.2 *
Type of core	Soft	280	41.5 a	−4.4 *	77	61.2 b	4.4 *
	Hard	382	58.5 a	4.4 *	52	38.8 b	−4.4 *
Padel racket weight (g)	<350	122	15.2 a	−6.0 *	53	36.1 b	6.0 *
	≥350	681	84.8 a	6.0 *	94	63.9 b	−6.0 *
Number of over grips	None or 1	443	55.2 a	−2.7 *	99	67.3 b	2.7 *
	2 or more	360	44.8 a	2.7 *	48	32.7 b	−2.7 *
Composition	Glass fiber	68	8.5 a	1.5	7	4.8 a	−1.5
	Carbon fiber	588	73.2 a	2.1 *	95	64.6 b	−2.1 *
	Unknown	147	18.3 a	−3.4 *	45	30.6 b	3.4 *

N: number; %: percentage; CSR: corrected standardized residuals; *: >|1.96|; a, b: indicate significant differences in Z tests comparing the column proportions, adjusting *p* < 0.05 according to Bonferroni.

**Table 3 ijerph-19-16858-t003:** Differences in the experience and volume of habitual padel practice between men and women.

Variables		Men			Women		
		N	%	CSR	N	%	CSR
Hours/week	<3 h	232	28.9 a	−2.9 *	60	40.8 b	2.9 *
	≥3 h	537	71.1 a	2.9 *	87	59.2 b	−2.9 *
Time practicing	<5 years	412	51.3 a	−2.7 *	93	63.3 b	2.7 *
	≥5 years	391	48.7 a	2.7 *	54	36.7 b	−2.7 *

N: number; %: percentage; CSR: corrected standardized residuals; *: >|1.96|; a, b: indicate significant differences in Z tests comparing the column proportions, adjusting *p* < 0.05 according to Bonferroni.

**Table 4 ijerph-19-16858-t004:** Differences in the characteristics of upper limb injuries between men’s and women’s amateur padel players.

Variables		Men			Women		
		N	%	CSR	N	%	CSR
Type of injury	Muscular	144	34.0 a	0.8	17	28.8 a	−0.8
	Ligamentous	43	10.2 a	−2.3 *	12	20.3 b	2.3 *
	Tendinous	206	48.7 a	0.7	26	44.1 a	−0.7
	Bone	30	7.1 a	0.1	4	6.8 a	−0.1
Location	Shoulder	143	33.8 a	−0.5	22	37.3 a	0.5
	Arm	18	4.3 a	−0.3	3	5.1 a	0.3
	Elbow	148	35.0 a	−0.1	21	35.6 a	0.1
	Forearm	50	11.8 a	0.0	7	11.8 a	0.0
	Wrist	53	12.5 a	1.3	4	6.8 a	−1.3
	Hand	11	2.6 a	−0.4	2	3.4 a	0.4

N: number; %: percentage; CSR: corrected standardized residuals; *: >|1.96|; a, b: indicate significant differences in Z tests comparing the column proportions, adjusting *p* < 0.05 according to Bonferroni.

**Table 5 ijerph-19-16858-t005:** Relationship between injury and racket characteristics in amateur padel players.

Variables		Injury			No Injury		
		N	%	CSR	N	%	CSR
Padel racket shape	Round	121	25.1 a	−1.7	140	29.9 a	1.7
	Tear	192	39.8 a	0.7	176	37.6 a	−0.7
	Diamond	169	35.1 a	0.8	152	32.5 a	−0.8
Type of core	Soft	202	41.9 a	−1.6	221	47.2 a	1.6
	Hard	280	58.1 a	1.6	247	52.8 a	−1.6
Padel racket weight (g)	<350	82	17.0 a	−2.3 *	107	22.9 b	2.3 *
	≥350	400	83.0 a	2.3 *	361	77.1 b	−2.3 *
Number of over grips	None or 1	263	54.6 a	−1.6	279	59.6 a	1.6
	2 or more	219	45.4 a	1.6	189	40.4 a	−1.6
Composition	Glass fiber	46	9.5 a	1.9	29	6.2 a	−1.9
	Carbon fiber	346	71.8 a	−0.1	337	72.0 a	0.1
	Unknown	90	18.7 a	−1.2	102	21.8 a	1.2

N: number; %: percentage; CSR: corrected standardized residuals; *: >|1.96|; a, b: indicate significant differences in Z tests comparing the column proportions, adjusting *p* < 0.05 according to Bonferroni.

**Table 6 ijerph-19-16858-t006:** Relationship between injury and padel practice (hours/week and time practicing) and time practicing another sport among amateur padel players.

Variables		Injury			No Injury		
		N	%	CSR	N	%	CSR
Hours/week	0–3 h	129	26.8 a	−2.7 *	163	34.8 b	2.7 *
	3–6 h	234	48.5 a	0.3	223	47.6 a	−0.3
	>6 h	119	24.7 a	2.7 *	82	17.5 b	−2.7 *
Time practicing	<5 years	289	60.0 a	2.6 *	241	51.5 b	−2.6 *
	≥5 years	193	40.0 a	−2.6 *	227	48.5 b	2.6 *
Play another sport	Yes	351	72.8 a	0.5	334	71.4 a	−0.5
	No	131	27.2 a	−0.5	134	28.6 a	0.5

N: number; %: percentage; CSR: corrected standardized residuals; *: >|1.96|; a, b: indicate significant differences in Z tests comparing the column proportions, adjusting *p* < 0.05 according to Bonferroni.

**Table 7 ijerph-19-16858-t007:** Relationship between the type of injury and its location in amateur padel players.

Men								
Location	Muscular		Ligamentous		Tendinous		Bone	
	%	CSR	%	CSR	%	CSR	%	CSR
Shoulder	41.7	2.5 *	55.8	3.2 *	24.8	−3.8 *	26.7	−0.9
Arm	6.3	1.5	0.0	−1.5	3.9	−0.4	3.3	−0.3
Elbow	22.2	−4.0 *	23.3	−1.7	48.5	5.7 *	20.0	−1.8
Forearm	22.2	4.8 *	4.7	−1.5	7.8	−2.5 *	0.0	−2.1 *
Wrist	5.6	−3.1 *	14.0	0.3	13.6	0.6	36.7	4.1 *
Hand	2.1	−0.5	2.3	−0.1	1.5	−1.4	13.3	3.8 *
**Women**								
**Location**	**Muscular**		**Ligamentous**		**Tendinous**		**Bone**	
	**%**	**CSR**	**%**	**CSR**	**%**	**CSR**	**%**	**CSR**
Shoulder	58.8	2.2 *	16.7	−1.7	38.5	0.2	0.0	−1.6
Arm	17.6	2.8 *	0.0	−0.9	0.0	−1.6	0.0	−0.5
Elbow	11.8	−2.4 *	66.7	2.5 *	38.5	0.4	25.0	−0.5
Forearm	11.8	0.0	16.7	0.6	11.5	−0.1	0.0	−0.8
Wrist	0.0	−1.3	0.0	−1.0	7.7	0.2	50.0	3.6 *
Hand	0.0	−0.9	0.0	−0.7	3.8	0.2	25.0	2.5 *

%: percentage; CSR; corrected standardized residuals; *: >|1.96|.

## Data Availability

Not applicable.

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
