# Peer review of "Incidence of Upper Body Injuries in Amateur Padel Players"

_ijerph, 2022, doi:10.3390/ijerph192416858_

Round 1

Reviewer 1 Report

1)      If p=0.000, please write it as p<0.001.

2)      Which Table is referred to the objective 1 “incidence (%) of upper body injuries?

3)      Why don’t you use chi-square test to answer your objective 2-4?

4)      Table 2-4, “a, b” significant of gender in Round, tear diamond? The table values are a bit confusing to readers I think.

5)      Should it be prevalence rather then incidence? If you are doing prospective study, then you should use “incidence”, if you are doing cross sectional study, it should be prevalence.

6)      What is the study design? Any sampling method used in selecting the participants?

7)      Any inclusion and exclusion criteria for participants?

Author Response

All the authors greatly appreciate the value they give to our work. Your comments and suggestions are very accurate and we will take them into account.

The manuscript has been reviewed by an English expert. We can add the revision certificate if necessary.

1) We have written p<0.001. As you have indicated

2) Table 4

3) In the results section, the chi-square test appears in all paragraphs before each table. Used in all analyses.

4) "a" and "b" are post Z tests that were performed to compare column proportions, adjusting p values ​​< 0.05 according to Bonferroni. It is indicated in the legend of each table.

5) We believe that "incidence" is more accurate since we are analyzing the appearance of injuries and not the prevalence of injuries.

6) We have added a section indicating the study design as indicated. No sampling method was used.

7) There were no inclusion or exclusion criteria for the participants. Just being amateur padel players

Reviewer 2 Report

This aimed to investigated the the injuries suffered during the previous year by padel players according to the characteristics of the racket, their usual volume of practice and their experience in padel. The topic of the study sound interesting, so I have just minor comments before suggesting it for publication.

What is the study hyphotesis?

Injury related to the sport can be associated with some anatomical aspects. If possible, could the author correlates injury incidence with limb length?

Were injuries associated to participants’ joint flexibility?

Were the participants engaged in resistance exercise program? Could the author the physical levels of the participants?  

Author Response

All authors appreciate the value they give to our work. Your comments and suggestions are very accurate and we will take them into account for future studies.

- This study did not collect the length of the limbs of the players.

- This study did not collect the joint flexibility or the physical level of the participating amateur padel players.

- We have decided not to include any hypotheses, since these statistical analyzes have never been carried out (for example: the relationships between racket characteristics and the incidence of injuries), so they had no idea of the results they were going to obtain. to get.

Round 2

Reviewer 1 Report

I have no further comment.